# Severe Loneliness and Isolation in Nursing Students during COVID-19 Lockdown: A Phenomenological Study

**DOI:** 10.3390/healthcare12010019

**Published:** 2023-12-21

**Authors:** Pingting Zhu, Wen Wang, Meiyan Qian, Guanghui Shi, Qianqian Zhang, Ting Xu, Huiwen Xu, Hui Zhang, Xinyue Gu, Yinwen Ding, Amanda Lee, Mark Hayter

**Affiliations:** 1School of Nursing, School of Public Health, Yangzhou University, Yangzhou 225009, China; mz120211876@stu.yzu.edu.cn (W.W.); mx120211164@stu.yzu.edu.cn (M.Q.); mz120211893@stu.yzu.edu.cn (G.S.); mz220210500@stu.yzu.edu.cn (Q.Z.); mx120201102@yzu.edu.cn (T.X.); 006721@yzu.edu.cn (H.X.); mx120221190@stu.yzu.edu.cn (H.Z.); mx120221199@stu.yzu.edu.cn (X.G.); mz120221984@stu.yzu.edu.cn (Y.D.); 2Jiangsu Key Laboratory of Zoonosis, Yangzhou 225009, China; 3Faculty of Health & Education, Manchester Metropolitan University, Manchester M15 6GX, UK; amanda.lee@mmu.ac.uk (A.L.); m.hayter@mmu.ac.uk (M.H.)

**Keywords:** COVID-19, nursing students, mental health, lockdown, loneliness, phenomenological study

## Abstract

In 2022, COVID-19 continued to spread across the globe, and to stop the spread of the virus and protect people’s health, universities across China continued to remain in a lockdown state. Loneliness is an important topic among college students, and the coronavirus pandemic has exacerbated loneliness. This prolonged school lockdown was unprecedented and it caused severe social isolation and emotional loneliness for students. Few people know how nursing students experience loneliness and find a way through their experience. This qualitative phenomenological study was conducted to reveal the lived experiences of nursing students who indicated COVID-19 lockdown-related loneliness in a previous quantitative survey. We performed 20 semi-structured interviews with nursing students aged 19–23 yrs during their lockdown (April 2022 to June 2022). Our research applied Colaizzi’s seven-step data analysis processes to reveal shared patterns in terms of how nursing students experienced lockdown and found the following four themes: emotional challenges associated with loneliness; causes of loneliness; positive and negative motivation to learn; and accepting solitude and reconstructing real life.

## 1. Introduction

Loneliness is a prevalent and common problem throughout the life course and is defined as a negative emotional experience that occurs when someone perceives a discrepancy between their actual social relationships and the social relationships they want, leading to impaired mental health [1]. This discrepancy may be related to the quality or quantity of the relationships [2]. Generally speaking, emotional loneliness is considered to be the absence of an attachment figure, while social loneliness is the absence of a social network, i.e., a circle of people that allows an individual to develop a sense of belonging, companionship, and to be part of a community [3]. Loneliness is associated with negative mental and physical health outcomes, including increased morbidity and mortality [4].

The coronavirus (COVID-19) outbreak began at the end of 2019 and spread immediately all over the world [5]. There have since been further variants of this severe acute respiratory syndrome [6]. One defense mechanism, (adjunct to the immunization strategy), was to adopt strategies to minimize person-to-person contact and therefore, human spread. The Omicron variant has been identified as more contagious than the original COVID-19 [7]. Although many countries have cancelled their contact prevention and control policies, China still adheres to dynamic zeroing measures (lockdown) [8]. The country’s epidemic responses (lockdown/immunization/antibody testing) have been implemented through a “national COVID-19 prevention and control plan” [9]. As a result, society has become increasingly restrictive, especially for young people, and this deleteriously affects mental health and wellbeing [10] as schools continue to impose multiple layers of restrictions on students’ movement. 

Nursing students, in particular, require clinical placements and have therefore been encouraged to pause their studies, which limits contact with people outside of the school. Existing students’ movements tend to be limited to two points of residence and the canteen. This isolation has become a major source of distress and has been linked with subsequent mental health and wellbeing decline in many studies [10,11,12].

These COVID-19 restrictions pose considerable emotional challenges such as feelings of isolation, stress, low mood, anxiety, and depression [13,14]. The mental health impact of living through a pandemic are very well-evidenced across the general population, especially in university and college students [15,16]. Though the risk of serious health implications following infection is lower in younger populations (more likely to be in full-time education), they are still required to adhere to lockdown measures to prevent spread to older populations [17]. 

Several studies on university students have linked lockdown measures with loneliness, and nursing students are no exception [18,19]. To complete their training, student nurses must fulfill clinical practice placements, delivering care to populations with a potentially higher risk of the disease, yet lockdown measures prevent this, and any delays in their training and education potentially cause a negative impact on mental health and wellbeing [1,10]. 

Our quantitative survey of student nurse experiences during lockdown revealed that a significant number of students expressed feelings of loneliness. This study identified a gap in our research and uncovered a need to understand the phenomena of loneliness in student nurses. Understanding their journeys may facilitate more effective support and management interventions that best support students throughout their nurse education and thus, produce a workforce ready for the demands of healthcare and public services. 

Phenomenology offers a qualitative method which seeks to understand, delve into these experiences, and to produce descriptions which reflect a truthful participant experience. We drew from the founding researchers in descriptive phenomenology—Colaizzi (1978) proposed a seven-step process, which provides an all-encompassing description of a phenomenon, which precisely fits with the overall aim of our study—to understand the feelings of loneliness and isolation in student nurses during their lockdown experiences. The objective of our paper is to use these experiences to guide the most appropriate support for future students’ mental health and wellbeing. 

## 2. Methods

### 2.1. Study Design

Our research applied Colaizzi’s 7-step phenomenological design to reveal shared patterns in terms of how nursing students experienced lockdown [20]. Using semi-structured interview questions (Table 1), we explored feelings and management of loneliness and isolation. The consolidated criteria for reporting qualitative (COREQ) research were used to ensure the transparency and rigorous reporting of our findings [21,22].

### 2.2. Participants and Settings

This study draws from a purposive sample of students (those who had already expressed ideations of loneliness and isolation in our previous study). We undertook semi-structured interviews to delve into their experiences until we reached data saturation [23], and a population deemed as representative of the total nursing cohorts across the country (N20). For a representative sample, we attempted to generate data from participants with age ranges of 19–23 and a ratio of 5:1 female: male (Table 2). 

Inclusion criteria consisted of the following: Any nursing student (Year 1–4) who had stated in our previous study that they had experienced loneliness. Students who were able to participate in, face-to-face, an interview on-site at Yangzhou University. Exclusion criteria consisted of the following: Students with pre-existing mental health/psychological concerns. Any students who feel they may be distressed about talking about their experiences of lockdown. 

### 2.3. Data Collection

Our interview team consisted of one experienced interviewer (PTZ), using semi-structured interview questions developed by highly experienced qualitative nursing researchers (Table 1) (MH, AL). Data collection took place between April and June 2022, in an online conference room. No one, other than the participant and researcher, was present during the interviews. The participants (all nursing students) were informed of the purpose of the study and assured of their confidentiality. They were assured that any responses would not impact on any subsequent learning experiences. Those consenting to participate in these audio-recorded interviews were invited to take part. The interviews lasted between 60 and 90 min. Open-ended questions were posed to consenting participants only and no participants withdrew from the study during or after their semi-structured interviews. 

### 2.4. Data Analysis

Interviews were transcribed and participants were asked to verify that the transcribed data was reflective of their thoughts and feelings [24]. The post-transcription data was analyzed independently so researchers could familiarize themselves with data. Coding was used to generate sub-themes and units of meaning in line with Colazzis’ [20] seven-step process (familiarization with data (all the research team read through transcribed data a number of times); identification of significant statements (identifying all statements which were of direct relevance to loneliness and isolation); forming meaning (bracketing pre-suppositions by discussions across the research team and considering any intrinsic/extrinsic influencers); clustering themes (clustering the themes and bracketing to account for any pre-existing knowledge on nurses’ experiences which may be derived from existing literature); developing a description which is exhaustive (documentation and reflection on all themes to ensure they capture all of the information gleaned at step 4); producing a fundamental structure and seeking verification of that structure (revisiting participants to review whether they felt that the themes reflected their lived experiences)). 

### 2.5. Ethical Considerations

Ethical approval was granted by the Yangzhou University ethics committee [Approval number 20220100]. In March 2022, participants were offered a written information leaflet and fully informed of the study rationale and potential time required for interviews. They were offered the opportunity to participate with the understanding that refusal would in no way impact any subsequent nursing studies. Data were anonymized by the single interviewer using a unique identifier and transcribed interview notes were kept in a locked cabinet in line with the data protection policy. All participants were assured that their responses were confidential. 

### 2.6. Rigor

This study is validated through principles of reliability, transferability, consistency, and verifiability [25]. To enhance credibility and reliability, the research team used bracketing and field notes to discuss and highlight any preconceived ideas—or any potential power relations which may affect participant responses. Two participants were invited to review their transcribed data and verify accuracy of transcriptions. The themes generated were discussed with all 20 participants to ensure they were consistent with their lived experiences. The research team developed the initial semi-structured interview questions from existing literature on loneliness. Participants were asked if the questions were enough to yield specific information about their loneliness experiences, and the open-ended question of whether participants wanted to add any further information offered the opportunity to add further detail. 

## 3. Results

Twenty nursing students participated in this study. A total of four participants were in their first year of study, five were in year 2, seven in year 3, and four were in year 4. The mean age of the participants was 20 years (SD = 1.16). A total of 85% of the participants were women. The students’ characteristics are listed in Table 2.

As described in the data analysis section, Coliazzi’s [20] seven steps were taken to develop meaningful themes from a cluster of sub-themes generated from significant statements (Table 3). These themes were verified with participants to ensure they reflected their true, lived experiences. The results are presented in these themes and sub-themes. 

### 3.1. Emotional Challenges Associated with Loneliness

Theme 1 encompasses the emotional challenges raised by a number of participants. As the length of the blockade increased, the participants experienced several emotional challenges which they associated with feelings of loneliness and isolation. The participants linked their experiences of loneliness with complex emotions. They revealed conflicts and power struggles when having to conform to rules and regulations, they felt like they were going to break down emotionally, their mood was impaired and they experienced anger, depression, and difficulties in communication with peers, family, and friends. 

#### 3.1.1. Power

All the participants were expected to fulfill duties under a number of rules and the violation of these rules would result in penalties. The participants’ movements were restricted throughout the Omicron epidemic, and this exacerbated feelings of powerlessness—all factors which caused increased loneliness.
“In those days rules were rules and if they were broken, I would be disciplined and punished”. (N3)
“The school was very strict and we were not permitted to go home or have any contact with outside the school”. (N7)
“It was like I am locked up in a prison and the world outside was bustling”. (N11)

#### 3.1.2. Emotions

As the closure time increased, most participants reported experiencing mood swings and they reported feelings of emotional collapse—reporting their experiences as though lockdown would never end.
“At first closed management, I didn’t feel it. Slowly I started to break down and even wanted to take time off to go home, but it was not an easy task”. (N8)
“I think the hardest thing to endure emotionally is the feeling of loneliness, I guess, and then there’s that sense of powerlessness of not knowing when the closure will end”. (N15)

#### 3.1.3. Acceptance

Although the stress was evident in most of the participants’ experiences, some of the participants indicated that they were able to learn to accept their situation and some were able to reconstruct their experiences positively, although barriers still existed. However, other participants saw this as a compromise.
“We have to adapt to make the best of each situation you find yourself in”. (N3)
“I feel that this management is a blow to a gradual adaptation until finally unblocked”. (N20)

#### 3.1.4. Depression

In the context of constrained freedom, participants also expressed frustration within their descriptions of loneliness during physical distancing. This frustration arose as a result of the forced restriction of freedom as well as the strict regulation.
“When I heard about the closure, I was very happy at first that I didn’t have to go to a hospital placement. But I could only move around the campus, I couldn’t even order takeaways and the extreme management was starting to depress me”. (N6)
“I lost all my desires, including the desire to be alive, and my life was like breathing in the ocean, threatening to suffocate at any moment”. (N12)

#### 3.1.5. Inability to Talk

As lockdown continued, the participants noticed their feelings of loneliness became more painful and exacerbated negative ways of thinking and acting in daily life. The participants recognized that communication between classmates was significantly reduced, and they felt that although they were experiencing the same things, there were still barriers to empathy.
“I only get along with my housemates on a polite basis, I prefer to confide in them face to face like my former friends, but we are limited”. (N8)
“I just lie in bed every day and the thought of closure is endless. It’s hard for me, I just don’t want to move or talk to anyone either”. (N19)

#### 3.1.6. Sensitivity to Emotions and Anger

Limited physical space for activities as well as containment in geographical locations strained interpersonal relationships, as the participants described experiences of conflict with housemates and boyfriends, and although these may have been resolved, they still had the potential for a psychological impact.
“When we are close together in the hostel, we become sensitive and fractions”. (N2)
“I disclosed my experience of loneliness to my boyfriend, but he didn’t understand the feelings that come with closure, so we clashed”. (N9)

### 3.2. Causes of Loneliness

Theme 2 reveals information on the factors the participants linked with their loneliness experiences. They cited that loneliness was exacerbated by separation and losing connections with family and friends. They identified that limited face-to-face interaction caused issues in communication. They suggested they were bored at times and that the isolation from social activities limited their ability to empathize with those around, or to disclose feelings. 

#### 3.2.1. Separation and Losing Connections with Family and Friends

The experience of loneliness may have occurred when the participants felt that they were out of their normal routine and had turned their backs on their families. People think that someone may feel better in the presence of their parents and that their company is comfortable.
“My family had to work and I was alone at school with nothing to do and they couldn’t be there for me, so I felt abandoned by my family. Because my family didn’t live in the same place as me, I felt disconnected”. (N4)
“My best friend is in the same city as me, but we can’t see each other”. (N7)

#### 3.2.2. No Face-to-Face Interaction

The participants stated they were deprived of face-to-face interaction and physical contact, and reported this made them feel very isolated. When asked to describe their experiences further, they mentioned disadvantages which they had associated with online contact. The participants unanimously expressed a preference for face-to-face communication rather than through social networking software (WeChat V8.0.20), revealing that face-to-face communication was considered to be warm and emotionally interactive.
“Because I tend to prefer face-to-face communication, I don’t think sending WeChat can serve the purpose of emotional communication”. (N6)
“We often go to online classes, we can no longer go to part-time jobs with our friends, we can no longer go to various competitions and meet new people, and even our relatives can only be seen during the Chinese New Year”. (N10)

#### 3.2.3. Social Isolation 

Many of our participants linked social isolation with increased feelings of loneliness. They reported strong feelings of disconnection from family and friends for many students, and linked this to visitation restrictions and restrictions on socialization.
“After being closed to school for so long, I felt disconnected from society. I’m just standing on the sidelines looking at this other person, and the world gives me a sense of unreality”. (N2)
“It was like I was locked up in a prison, it wasn’t the place for me to study, the world outside was too busy”. (N14)
“I was very lonely. You couldn’t go to the other students’ dormitories and we couldn’t see our friends. And our families couldn’t come and visit, so it was very lonely”. (N19)

#### 3.2.4. Boredom

The participants reported that they initially enjoyed staying in the halls of residence due to the restrictions during the pandemic, but as the process became longer, they began to become bored doing the same things every day. They described how nothing changed in their daily lives and the boredom they experienced as a result of the monotony, not enjoying life, starting to lose the meaning of life, and the unprecedented days everyone was experiencing.
“I got tired of staying at school and I started to lose patience with other people more easily. I also feel more bored, which can lead to other emotional upsets, impatience, and sadness”. (N7)
“It felt like everything was on hold for that time”. (N16)

### 3.3. Positive and Negative Motivation to Learn

Theme 3 relates to the experiences reported about motivation, drive, and commitment to learn during lockdown. The participants were required to balance their learning with lockdown and health surveillance—but also alongside their own health status. Some of the participants expressed positive feelings about this, others negative, citing a more passive (or altered) learning status. Some felt that lockdown prevented them from learning, while others gained more free time.

#### 3.3.1. Burnout

Participants reported that the requirement to complete additional tasks (such as daily temperature monitoring) made them feel more ‘burned out’ and ‘stressed’ during their studies. Some also linked loneliness with their feelings of burnout.
“I was glued to my phone every day for fear of missing any punch cards or COVID-19 related notifications, and I simply didn’t have the heart to study anymore”. (N1)
“Because studying in the dormitory means reading a book, but you can’t read it. You sit there studying and you lie in bed playing, and the state of mind is the same, that very empty feeling”. (N10)
“Back then, my classes were usually in the afternoons and I slept in the mornings, and my whole being was particularly burnout, which I know can be a sign of lack of exercise, but I was stuck in a vicious cycle”. (N13)

#### 3.3.2. Lack of Motivation

The participants reported failing to stay alert and attentive in class and linked this to several factors, including the teacher’s inability to interact with the students, the varying degrees of stress, lack of competition between classmates, and headaches caused by staring at the screen for long periods.
“My overall learning status is particularly poor and I am unable to concentrate in the face of online classes. My independent learning skills are not very strong and I need external competitive stimulation…” (N1)
“I need supervision from my teacher… Although I am a college student, the online classes have completely let my guard down. I will reflect on why I wasted so much time in the day not studying”. (N5)

#### 3.3.3. More Time to Prepare for Exams

The lockdown also offered opportunities for participants and some reported positive experiences—they enjoyed the freedom to be masters of their own time.
“The freedom is more restricted at school, but there is more time to study, go to the library and study rooms, and go for a walk in the playground in the evenings if you feel bored”. (N = 20)
“The closed management was a slot and opportunity to improve me. I usually needed to go to hospital practice and had a lot of time to prepare for my exams, which I found quite enjoyable”. (N = 9)

#### 3.3.4. Reduced Focus to Study

Some of the participants claimed they were unable to focus on their studies because they were used to study groups, Practice-Based Learning (PBL), and practical classrooms to do so. Similarly, the participants felt they were being deprived of clinical experience with patients. To protect themselves from more painful mental suffering, they were often desperate for an outlet for pleasure and had to pay less attention to their studies.
“I was getting depressed; how could I be bothered to study”. (N2)
“For brief relief from loneliness, I would spend a lot of time entertaining rather than studying”.(N14)
“I basically didn’t participate much in clinical practice, and as a nursing student, I only administered an IV to a patient once, and I didn’t know how to get into a hospital and become a real nurse”. (N15)

### 3.4. Accepting Solitude and Reconstructing Real Life

Our final theme focusses on the adaptation phase, and reveals how participants managed their feelings, and found emotional outlets to combat their loneliness. It describes how they began to accept and manage their loneliness and their emotions, so they could continue with their real lives.

#### 3.4.1. Adapting

Solitude was described by the participants as a ‘healing’ process, where they quieted down and absorbed new energy to continue with their closed lives.
“I have to try to live peacefully with loneliness…” (N9)
“I tried to be alone, to be quiet and absorb new energy”. (13)

#### 3.4.2. Keeping Busy

The participants reported that creating a busy environment helped to reduce feelings of isolation. The participants were excited about activities that gave them purpose and helped them connect with something they enjoyed. They indicated distraction activities which were helpful in combatting feelings of loneliness and isolation, such as sports, online shopping, hobbies.
“I distract myself with shopping. When I’m particularly lonely, I use it as a distraction”. (N19)
“I made a plan for myself during the closure period, such as how much studying to do, and then I would give myself time to exercise in the evening”. (N3)

#### 3.4.3. Socialization after Lockdown Meeting Friends Again

Many participants recalled the free life they had led in the pre-pandemic period, highlighting that talking to friends and family about memories and good times was helpful. They expressed a longing for time spent in restaurants, cafes, face-to-face classes, and hospital practice in the pre-pandemic period, for the days when they walked freely in the streets, did sports and could attend concerts and movies. They also reported that they realized during the pandemic that every moment they had before the pandemic was very valuable. This yearning for the future was clearly described.
“My friend and I agreed to go shopping together after the epidemic. It’s the way to cure me”.(N20)
“After liberation, I went for a few KFC meals with my real-life friends”.(N17)

#### 3.4.4. Interaction of Digital Network

Technology was reported favorably as great resource for connecting geographically dispersed family, friends, and the wider world. The participants used technology in creative ways so that they could remain connected. Although the participants arranged online appointments and used the internet for contact with the outside world, and increased connections as a means to reduce the loneliness, many of the participants identified that this was not a permanent solution.
“I started webcasting and met a lot of people on social networks with whom I felt I could relate”.(N7)
“I am very happy to receive phone calls or video calls from my family every day, and I am in closer contact with them on WeChat”.(N12)

## 4. Discussion

This study aimed to understand feelings of loneliness and isolation in student nurses during their lockdown experiences. The qualitative analysis of the semi-structured interviews with nursing students at Yangzhou University in Jiangsu Province produced four themes: (1) emotional challenges associated with loneliness, (2) causes of loneliness, (3) positive and negative motivation to learn, (4) accepting solitude and reconstructing real life. 

Measures to contain the spread of the COVID-19 virus have resulted in school closures and have curtailed young people’s social contacts outside of their immediate household. The subsequent disruption in education, transition to distance education, uncertainty of experience, disruptions across clinical practice, and first encounter with a pandemic has negatively impacted students who were still adjusting to the profession and university [26] and our study sought to delve into the lived experiences of these students. 

Our findings support existing studies linking a steep reduction in social contact with increased experiences of loneliness and potential psychological distress [10,27]. The loneliness experienced by our student groups occurred at a particularly important time in their lives—as study commitments, separation from families, and lack of income (because of studies) can exacerbate these feelings, which is supported by [1]. We identified that feelings of loneliness were indicated throughout the whole of the participants’ experiences and that these feelings were exacerbated by a number of factors. One study suggests this can overshadow and/or undermine the ‘student experience’, adding to the pressures to perform and exacerbating the anxiety experienced [28].

As a result of the compulsory lockdown, our participants reported a constantly changing mental health status—expressing fear/anger/powerlessness and an inability to communicate effectively—this aligns with a study which identified loneliness and other emotional challenges are constantly in fluctuation, with complex emotions occurring over a long time period during lockdown [29]. 

Our study participants linked these feelings of powerlessness with frustration, distress, and loneliness, as they were required to conform to the social isolation of lockdown measures. As restrictions forced housemates to spend a lot of time together, they reported increasingly strained relationships. A recent study conducted by Kaur et al. also supports this claim. Their study of Polish students during lockdown linked isolation with increased loneliness and other negative emotions [12].

The objective of our paper was to use the understanding of our participants’ experiences to guide the most appropriate support mechanisms which would guide and assist future students’ mental health and wellbeing. Thus, we can conclude the most relevant factors linked with abating loneliness was to encourage family and friends’ support. This is further indicated by findings from a study [30] that also calls for students to never hesitate to seek help.

This support should be face-to-face as far as possible—because a lack of contact and limited social interaction can lead to psychological and emotional loss for some people [31]. Our results illustrated how the internet is no substitute for face-to-face social interaction; it merely acts as a temporary outlet. Our findings are supported by recent experimental studies that suggest digitally provided social support does not improve mood when compared to face-to-face support [32]. Emotional connections, and combatting boredom through social interaction is key to overcoming the issues of loneliness, and all our participants noted this was the case. A study also links loneliness with physical confinement, estrangement from loved ones, and psychological distress caused by a lack of emotional and social connections [33]. The factors attributed to this in Hwang’s study included an inability to synchronize life pace—and this may be attributable to our findings—in which our participants used ‘staying busy’ as a mechanism for coping with boredom and stress [33]. 

As we move to adaptive mechanisms which could improve our student experiences, and use findings from this paper to support better initiatives which would potentially combat loneliness, we also need to consider the additional tasks associated with a health lockdown. Di Pietro’s study evaluated the impact of health-related ‘tasks’ on academic studies, and revealed they caused increased frustration and fatigue [34]. Our participants also indicated this was problematic. Di Pietro calls for more engaging online learning methods to be used during confinement, as a means to combat a lack of motivation, to increase attention to learning, and reduce any potential academic burnout [34]. 

Of course, there are positive factors associated with lockdown, and our participants identified more time to focus and prepare for exams. Our results are consistent with previous studies conducted during the COVID-19 pandemic; for example, a study identified some of their participants felt that they had a great deal of free time and a better balance between study and life during lockdown [35].

Our fourth theme perhaps captured the factors most relevant to our overall objective of the study. It indicated mechanisms which support students to manage their experiences. While stress was evident in most of our participants, some reported coping strategies which facilitated them to reshape their reality. They indicated socialization—hobbies and face to face contact—as a means to cope. Taylor’s study supports this, and emphasizes the importance of enjoying free and leisure activities, taking physical exercise, connecting with friends and family, and engaging in a busy schedule of study as a means to combat loneliness [36]. 

Although the internet offers a great resource for connecting geographically dispersed family, friends, and the wider world, our participants reported that it did not compare with face-to-face contact. People can use technology in creative ways to stay connected [37], and many of our participants reported that they had arranged videos with family, played games with friends, etc. The fact remains that although digital connection is a key mechanism for socially connecting during the pandemic, it is only considered as a means to an end—and is no substitute for face-to-face interaction [38,39].

## 5. Limitations

As a single center study, our results may only be representative of our students. However, the themes generated appear to be reflected across the literature—and illustrate coping strategies which are relevant to a number of wider areas. Although the staff were very clear to identify to the participants that the study would not affect their marks, or any subsequent study, we realize that there may have been some element of biased response. However, as the students were only discussing their experiences of loneliness in context, and nothing to do with their ability or commitment to studies, this was felt to be a reasonable request. The team focused heavily on confidentiality and there were many discussions on bracketing across the research team. The population sample was drawn from a wider sample quantitative study, and we chose only participants who had expressed feelings of loneliness. With this in mind, we may have missed some students who had feelings of loneliness, but did not express this in our quantitative study. However, we mitigated this risk by ensuring interviews were completed only on data saturation. 

## 6. Conclusions

The study revealed the feelings of loneliness and isolation in student nurses during their lockdown experiences. Our themes represented the student’s experiences and identified some factors which they found useful to combat loneliness. 

The findings showed that the students experienced a range of emotional challenges and feelings of isolation triggered by social isolation, and that the university being closed also brought positive or negative aspects of learning. The internet is no substitute for face-to-face social interaction, and we need to find more innovative and engaging solutions to support our students in their learning journey. 

Allowing time for social contact is essential for our student’s mental health status as a means to combat the feelings of isolation and loneliness during and after lockdown. Nursing education needs to offer more engaging provisions to develop safe spaces for students who may be ‘losing focus’ or having difficulties in their attention where they can discuss these feelings with support teams or wellbeing centers, who could offer support and advice to students—to offer information which links students to social networking and activities. 

Nursing academics need to realize the importance of socialization as a means to combat loneliness. They need to offer support and safe spaces for students to reveal feelings of loneliness and identify potential stressors. 

## Figures and Tables

**Table 1 healthcare-12-00019-t001:** Semi-structured interview guide.

Interview Guide
(1) Can you tell us how you coped emotionally and mentally with the lockdown measures during the COVID-19 pandemic?
(2) How did you manage when you felt lonely?
(3) Did you ever ask for help with these feelings?
(4) Who did you contact for help?
(5) Did you use any other means of communicating with your friends/relatives during the lockdown?
(6) How would you rate yourself in terms of personality?
(7) Are you generally an outgoing person? Do you usually seek help from friends and family for social/emotional support?
(8) Finally—what advice would you give to new students on the best ways of coping—if a lockdown were to happen again. ADDITIONAL—is there anything more you would like to say about your experience?

**Table 2 healthcare-12-00019-t002:** Participant characteristics.

n = 20	n (%)
Course year	
First year	4 (20%)
Second year	5 (25%)
Third year	7 (35%)
Fourth year	4 (20%)
Gender	
Female	17 (85%)
Male	3 (15%)
Age	
<19 yrs	3 (15%)
19–20 yrs	9 (45%)
21–23 yrs	8 (40%)

**Table 3 healthcare-12-00019-t003:** Themes and sub-themes derived from Coliazzi’s model of analysis of phenomenological data (Coliazzi, 1978) [20].

Themes	Sub-Themes
Emotional challenges associated with loneliness: loneliness experienced along with other negative emotions at school lockdown.	(1) Power (2) Emotions (3) Acceptance (4) Depression (5) Inability to talk (6) Sensitivity to emotions and anger
Causes of loneliness: these things lead to a sense of isolation for the participating students.	(1) Separation and losing connections with family and friends (2) No face-to-face interaction (3) Social isolation (4) Boredom
Positive and negative motivation to learn: closed management can be a double-edged sword for students’ learning.	(1) Burnout (2) Lack of motivation (3) More time to prepare for exams (4) Reduced focus to study
Accepting solitude and reconstructing real life: coping strategies that aim to alleviate the negative emotional impact of loneliness.	(1) Adapting (2) Keeping busy (3) Socialization after lockdown, meeting friends again (4) Interaction of digital network

## Data Availability

The data presented in this study are available upon reasonable request from the authors. The data are not publicly available due to privacy or ethical restrictions.

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
