# Peer review of "Severe Loneliness and Isolation in Nursing Students during COVID-19 Lockdown: A Phenomenological Study"

_healthcare, 2023, doi:10.3390/healthcare12010019_

Round 1

Reviewer 1 Report

Comments and Suggestions for Authors

GENERAL

To improve the flow and presentation of your manuscript consider using the APA manual. American Psychological Association. (2020). Publication manual of the American Psychological Association (7th ed.). Washington, DC: American Psychological Association. A target journal may not follow the APA style but the APA manual can help improve manuscripts greatly.

The manuscript must be proofread as grammar (e.g., tense and wording) has to be closely examined. For example in line 81 you say that “participants will be” but as the study has already been conducted the past tense should be used.

ABSTRACT

Lines 15-26. This is a time (history) sensitive study. Thus you need to give the reader a clear idea when data was gathered, year(s) and month(s), and what restrictions were in place when data was gathered.

Line 22. Consider reporting the gender split and age range (min-max) here.

Line 24-26. Your last sentence in the abstract suggest that you are going to list the four themes and 18 sub-themes but then you only list the four themes and not the 18 sub-themes.

INTRODUCTION

Line 44. When you talk about “only known defence” being social isolation why would you ignore vaccines? If you are discussing the COVID-19 situation at the start of the outbreak then yes social isolation may have been the only defence but you are not stating this clearly in this section even though you start by talking about the start of the outbreak. Please rethink the start of the paragraph starting in line 41 maybe by giving the reader a clear but concise idea about how these students would have been affected at different times from 2019 to 2022.

You need to make sure that you cover the key concepts in the introduction. You only mention the key term “mental health” once in the introduction, another key term “lockdown” is not mentioned at all in the introduction. The introduction needs to cover all of your key concepts, integrating them into a cohesive argument that is followed by clear and concise hypotheses at the end of the introduction. The hypotheses should then be clearly and systematically tested in the results section followed by a clear assessment in the discussion section.

Line 70. You are not testing “targeted psychological intervention strategies” so this should be dropped from the manuscript. You might be able to mention such strategies in the discussion but that would have to be done carefully.

Make sure that it is clear to the reader why this study is needed.

Line 81. Note comment on tense above.

METHOD

Section 2.2.

Give the reader a better idea about your sample here. Briefly report the number of males and females in your sample and the mean age and age range.

Section 2.3

Give the reader a clear idea about dropout rate(s). How many were offered an interview, if known, and of the ones that accepted the offer how many carried through with the interview (e.g., any dropout during interview?).

Section 2.5

Report the ethics code/number for your study, if applicable.

RESULTS

Line 141. Table 3 is provided but never referred (cited) in text. You need to talk about how the theme and sub-theme results were arrived at. Help the reader understand how you go from applying Colaizzi’s design to your themes and sub-themes as presented in Table 3.

The caption for Table 3 needs to be more descriptive (consult the APA manual).

Watch for inconsistency. In line 140 you say “sub themes” without a hyphen, however, in the table the column header says “sub-themes” with a hyphen.

Line 143. It would be good to put this into a better ‘timeline’ perspective for the reader. When (year and month) is the blockade increasing. This might be fixed if addressed earlier (e.g., methods) see previous comment as well.

Line 284. PBL is only mentioned once so rather than using the abbreviation consider just spelling it out.

DISCUSSION

Make sure that it is clear to the reader what the outcome was for the different hypotheses and how your findings relate to findings from previous studies.

Make sure to avoid introducing new citations in the discussion section unless necessary. If you do, check if you should not have included these new citations in the introduction. With this in mind note Tasso et al. and Barari et al. along with other new citations.

Line 338. Why are there two ** as in “**University in …”?

Line 354. You introduce a new concept here “psycho-spiritual state”. Suggest you consider introducing this concept earlier or explain to the reader how you derive it from loneliness and other emotional challenges as this is the first time anything spiritual is mentioned in the manuscript.

Lines 365-369. You are introducing new concepts and ideas here. Avoid extending your discussion (e.g., anxiety, depression, techniques, emotional support) too far from the actual focus of your study.

Lines 370-403. Consider if some of this text should not first appear in the introduction section to help justify your hypotheses. If you want to use it in the discussion section make sure to connect it clearly to your findings.

Lines 404-409. Consider merging this section in with the conclusion section.

CONCLUSION

Make sure to focus on the findings from your study and described in your manuscript and not to extend your conclusions too far beyond your findings.

Comments on the Quality of English Language

Please fix language as needed, note comments to authors. 

Author Response

Dear reviewer,

Reviewer 2 Report

Comments and Suggestions for Authors

Congratulations on the topic investigated, given its importance in the sample studied, as it is the nursing professionals who will later care for the rest of the population.

The qualitative methodology applied in this design is rigorous, however there are aspects that could further improve the article.

Method

In the inclusion of the subjects, it was considered that they stated that they felt lonely, but which professional assessed this feeling of loneliness? with what scale?

Is it necessary to include the 2007 Tong et al. reference when we have a more up-to-date reference from 2020?

I have not observed the code of the ethics committee of the institution.

Results

Perhaps it would have been interesting to stratify the sample (number of participants) according to age, being of the same proportion.

Check whether the capital N. This refers to the population (all subjects), but the lower-case n is the sample. For example, in table 2.

Discussion

Assess whether it is possible to remove the 2017 reference and replace it with a more up to date one. Only articles from the last 5 years should be used in the discussion.

There are no references from 2022 and 2023. Surely there are publications. It would be good for the article to look for references from those two years. Science in 2 years may have changed a lot.

Limitations

Limitations, although a priori not such an important section, should be more specific. It is not enough to say that they are those inherent to the design of the study, in this case of a qualitative type, but rather that within this study, what limitations have been found. The same applies to the statement that the research has been carried out in a specific geographical area or institution: in this case, the title of the article should appear, as it should not be generalised.

Author Response

Dear reviewer,

Round 2

Reviewer 1 Report

Comments and Suggestions for Authors

You need to check what style of citation you should use for this journal. The citations are not numbered correctly. First citation is [33] in line 35.

Line 161. Suggest you change the first column header. Change “n=20” to “Characteristics”. Also change the second column header. Change “no (%)” to “n (%)”.

Lines such as 51, 397, and 402. You need to use the numbered citation style for this journal.

Should the conclusion section not come after the limitation section? Check journal guidelines and convention. 

Comments on the Quality of English Language

NA
